# Platicon microcomb generation using laser self-injection locking

Grigory Lihachev[1,4], Wenle Weng [1,3,4], Junqiu Liu [1,4], Lin Chang [2], Joel Guo [2], Jijun He[1], Rui Ning Wang [1], Miles H. Anderson[1], Yang Liu [1], John E. Bowers [2] & Tobias J. Kippenberg [1✉]

The past decade has witnessed major advances in the development and system-level applications of photonic integrated microcombs, that are coherent, broadband optical frequency combs with repetition rates in the millimeter-wave to terahertz domain. Most of these advances are based on harnessing of dissipative Kerr solitons (DKS) in microresonators with anomalous group velocity dispersion (GVD). However, microcombs can also be generated with normal GVD using localized structures that are referred to as dark pulses, switching waves or platicons. Compared with DKS microcombs that require specific designs and fabrication techniques for dispersion engineering, platicon microcombs can be readily built using CMOS-compatible platforms such as thin-film (i.e., thickness below 300 nm) silicon nitride with normal GVD. Here, we use laser self-injection locking to demonstrate a fully integrated platicon microcomb operating at a microwave K-band repetition rate. A distributed feedback (DFB) laser edge-coupled to a $Si_3N_4$ chip is self-injection-locked to a high-$Q$ ($> 10^7$) microresonator with high confinement waveguides, and directly excites platicons without sophisticated active control. We demonstrate multi-platicon states and switching, perform optical feedback phase study and characterize the phase noise of the K-band platicon repetition rate and the pump laser. Laser self-injection-locked platicons could facilitate the wide adoption of microcombs as a building block in photonic integrated circuits via commercial foundry service.

[1] Institute of Physics, Swiss Federal Institute of Technology Lausanne (EPFL), CH-1015 Lausanne, Switzerland. [2] ECE Department, University of California Santa Barbara, Santa Barbara, CA 93106, USA. [3] Present address: Institute for Photonics and Advanced Sensing (IPAS), and School of Physical Sciences, The University of Adelaide, Adelaide, SA 5005, Australia. [4] These authors contributed equally: Grigory Lihachev, Wenle Weng, Junqiu Liu. ✉email: tobias.kippenberg@epfl.ch

In recent years, microcombs[1,2] have emerged as chip-scale optical frequency combs with repetition rates in the gigahertz-to-terahertz domain. The operation of DKS microcombs[3,4] has enabled many applications such as coherent communication[5,6], astronomical spectrometer calibration[7,8], massively parallel coherent LiDAR[9], optical and microwave frequency synthesizers[10–12], and photonic convolutional neural network[13,14]. Embracing the technological maturity of silicon photonics[15], today, fully integrated microcombs can be built on nonlinear photonic integrated circuits (PIC) fabricated using CMOS-compatible materials and processes. With its inherent material properties including the 5-eV bandgap (which leads to negligible two-photon absorption at telecommunication bands), the strong Kerr nonlinearity and the high power handling capability (more than 10-watt continuous-wave laser power on-chip[16]), $Si_3N_4$ has become the leading platform for integrated DKS microcombs[2,17]. Particularly, ultralow optical losses down to 1 dB/m and microresonator quality factor $Q > 10^7$ have been routinely achieved in $Si_3N_4$[18–21].

As anomalous GVD is mandatory for DKS formation, dispersion engineering via waveguide geometry variations[22,23] is required to overcome $Si_3N_4$'s normal material GVD. Anomalous GVD necessitates $Si_3N_4$ film thickness exceeding 600 nm at telecommunication bands. However, when using low-pressure chemical vapor deposition (LPCVD) that offers $Si_3N_4$ films of superior quality in terms of density, homogeneity, uniformity, and stoichiometry, film cracks often occur when the film thickness exceeds 500 nm. To prevent crack formation, multiple schemes have been demonstrated, including using pre-structured substrates[24,25] and depositing $Si_3N_4$ in multiple cycles[26–28]. However, these methods have not yet been incorporated into most of commercial $Si_3N_4$ foundry processes, where LPCVD $Si_3N_4$ films with thickness below 400 nm are offered[29]. Alternatively, plasma-enhanced chemical vapor deposition (PECVD) $Si_3N_4$[30–33] and silicon-rich LPCVD nitride[34–36] are free from crack formation. However, these $Si_3N_4$ films have not obtained ultralow losses comparable to those achieved with LPCVD stoichiometric $Si_3N_4$. Consequently, the wide adoption of $Si_3N_4$ DKS microcombs as a building block in standard PIC architectures via foundry service has not been possible.

Very promising in this context is that coherent optical pulse structures which are referred to as dark pulses[37–39], switching waves[40] or platicons[41], can be generated in microresonators with normal GVD. Compared with DKS, platicon generation not only relieves the effort for dispersion engineering, but also exhibits a larger pulse duty cycle and a higher comb conversion efficiency[39]. Despite these advantages, platicons are usually more difficult to obtain with the conventional frequency sweeping of a continuous-wave (CW) pump laser. This is due to the dynamically stable regime of platicons in the pump-resonance detuning is narrow and can be practically unstable[42] with a dispersion landscape dominated by the second-order coefficient (i.e. GVD). As a result, a majority of previous works focused on adopting pump modulation schemes[43–46] or using mode-coupling-based local dispersion alteration[37,47–49] to initiate platicon formation. These schemes either increase the experimental complexity, or reduce the reliability and the consistency of the performance.

In this work, we demonstrate a stable platicon microcomb formed in a normal-GVD $Si_3N_4$ microresonator by using laser self-injection locking. Originally used to create narrow-linewidth lasers[50–54], self-injection locking also offers routes to build compact microcombs with crystalline whispering-gallery-mode resonators[55,56] or chip-scale integrated microresonators[57–62]. Since self-injection locking couples two nonlinear systems—the semiconductor laser and the Kerr-nonlinear microresonator—to reach a dynamic equilibrium that is only accessible via self-locking mechanism, the platicon regime can be readily accessed

and firmly sustained[63,64]. Such a stable operation is demonstrated in our experiment with the measurements of platicon phase noise and second-order autocorrelation. Additionally, we numerically model the joint nonlinear system by coupling two theoretical models, namely the laser rate equations for the self-injected semiconductor laser[65,66] and the Lugiato–Lefever equation for the Kerr microresonator[67,68], showing stable platicon generation with self-injection-locked laser that is in agreement with our experimental observations.

## Results

### Characterization of photonic chip-based, high-Q, $Si_3N_4$ microresonators

Figure 1a presents the operational scheme of our experiments. A commercial DFB laser diode with 1556 nm center wavelength and output power up to 100 mW is self-injection-locked to a $Si_3N_4$ microresonator resonance. The locking is triggered owing to the Rayleigh backscattering[69] in the microresonator. Through direct edge-coupling with an engineered inverse taper[70] (cf. Fig. 1c), laser power up to 20 mW is delivered to the $Si_3N_4$ bus waveguide that is side-coupled to the microresonator. The microresonator shown in Fig. 1b has a geometry of 2.00 $\mu$m waveguide width, 530 nm height, and 900 $\mu$m ring radius, which leads to a normal GVD. We note that this waveguide geometry offers strong optical confinement, allowing for the generation of platicon microcombs with repetition rates in the millimeter-wave domain[34,37] and the microwave domain (this work). Frequency-comb-assisted diode laser spectroscopy[71] is used to characterize the microresonator dispersion profile and resonance linewidths in the telecommunication band from 1500 to 1630 nm. Here we focus on the fundamental transverse-electric mode (TE$_{00}$), as the edge-coupled DFB laser beam is TE-polarized. Figure 1d (top panel) shows the measured integrated microresonator dispersion defined as $D_{int}(\mu) = \omega_\mu - \omega_0 - \mu D_1 = D_2\mu^2/2 + D_3\mu^3/6 + D_4\mu^4/24 + \ldots$, where $\omega_\mu$ is the angular frequency of the $\mu^{th}$ resonance, $\omega_0/2\pi = 192.62$ THz is the frequency of the pumped resonance, $D_1/2\pi = 26.2$ GHz is the microresonator free spectral range (FSR), $D_2/2\pi = -59.9$ kHz is the GVD coefficient, and $D_3$ and $D_4$ are higher-order dispersion terms. The dispersion deviation from the $D_2$-dominant parabolic profile, defined as $[D_{int}(\mu) - D_2\mu^2/2]/2\pi$, is shown in the bottom panel of Fig. 1d. Figure 1e plots the histogram of the intrinsic linewidth $\kappa_0/2\pi$ of 2414 measured resonances. The microresonator is overcoupled, with the most probable $\kappa_0/2\pi = 19$ MHz, corresponding to an intrinsic quality factor of $Q_0 = 1 \times 10^7$. The measured back-reflection from the pumped resonance is ca. 15% in power, corresponding to a normalized forward-backward mode-coupling coefficient of $\beta = \delta/\kappa \approx 0.3$, where $\delta$ is the backscattering-induced coupling rate and $\kappa$ is the loaded loss rate (cf. Fig. 1f). More characterization results of the $Si_3N_4$ chip are presented in the Supplementary Information. Figure 1g shows a qualitative depiction of the laser frequency detuning trajectory when the laser current is linearly tuned. It can be observed that the laser frequency can be tightly locked to the microresonator resonance, although the laser bias current is varied within a certain range. We note that, for simplicity, in this plot, the microresonator resonance frequency is treated as fixed, i.e., the Kerr nonlinear frequency shift is neglected. We refer the readers to the recent work[59] that includes the microresonator Kerr frequency shift for a more accurate description of the laser frequency trajectory.

### Phase-noise characterization of the DFB laser

We characterize the DFB laser noise by beating it with a reference laser and analyze the single-sideband (SSB) phase noise of the beat signal using a fast photodetector and an electrical spectrum analyzer

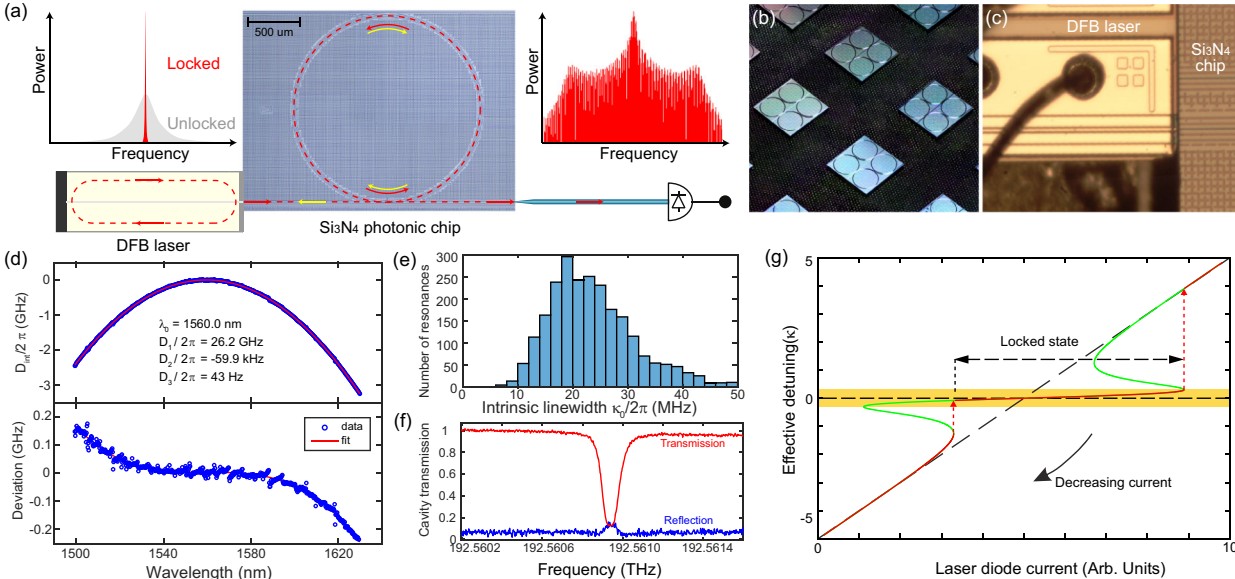

**Fig. 1 Principle of platicon microcomb generation using laser self-injection locking. a** Schematic of platicon generation using a DFB laser self-injection-locked to a photonic chip-based, high-$Q$, $Si_3N_4$ microresonator. The bulk and surface Rayleigh scattering inside the $Si_3N_4$ microresonator induces back-scattered light that is sent back to the DFB laser and triggers laser self-injection locking. This results in laser linewidth (i.e. frequency noise) reduction. Further increasing the laser power can trigger platicon formation due to the nonlinear interaction of counter-propagating cavity modes and modified laser dynamics due to the narrow-bandwidth Rayleigh backscattering. **b** Photograph of $Si_3N_4$ chips. **c** Photograph showing the DFB laser edge-coupled to a $Si_3N_4$ waveguide. **d** Top: Measured integrated dispersion of the $Si_3N_4$ microresonator. This microresonator has an FSR of 26.2 GHz in the microwave K-band and a normal GVD of $D_2/2\pi = -59.9$ kHz. Bottom: The dispersion deviation from the $D_2$-dominant parabolic profile reveals avoided mode crossings and higher-order dispersion terms. **e** Histogram of 2414 fitted intrinsic resonance linewidth $\kappa_0/2\pi$. The most probable value is $\kappa_0/2\pi = 19$ MHz, corresponding to an intrinsic $Q_0 = 10^7$. **f** Microresonator transmission (red) and reflection (blue) traces showing the resonance with 15% back-reflection. **g** Simplified analytical estimation of laser frequency tuning curve based on the model of linear self-injection-locking illustrated in ref. [59]).

(ESA). The DFB laser, whose optical spectrum is shown in Fig. 2a (inset), can either be in the free-running state or the self-injection-locked state. The operation state depends on the frequency detuning between the microresonator resonance and the DFB laser-cavity resonance that is controlled by the laser bias current. Figure 3a shows the schematic of the experimental setup. The phase noise measurement was performed using heterodyne technique with a reference laser. An external-cavity diode laser (ECDL, Toptica CTL) locked to an etalon Fabry–Pérot cavity of 30-kHz resonance bandwidth with Pound-Drever-Hall (PDH) technique is used as the reference laser to detect the DFB laser noise. Figure 2b shows the measured SSB phase noise spectra of the DFB laser (in the free-running or self-injection-locked states, respectively, with 370 mA laser bias current) and the reference laser. At Fourier offset frequencies above 300 kHz, our measurement is limited by the servo bump from the PDH lock bandwidth. Compared to the free-running state, more than 30 dB phase noise reduction is observed at Fourier offset frequencies up to 1 MHz when the laser is self-injection-locked. Similar phase noise reduction has also been obtained with self-injection locking using anomalous-GVD $Si_3N_4$ microresonators[72]. We expect further laser phase noise reduction with an optimized optical feedback phase and the full packaging of the system. In addition, using microresonators with large mode volumes[73,74] that mitigate the impact of fundamental thermo-refractive noise[75] could be beneficial to further reduce the laser's intrinsic linewidth.

**Platicon microcomb generation and characterization.** As shown in Fig. 2a, with the laser being self-injection-locked, we can generate a single-FSR-spacing (26.2 GHz) platicon microcomb, by judiciously setting the DFB laser temperature and the bias

current. We observe spectral asymmetry and comb line power variation, which can be attributed to the influence of higher-order dispersion terms and the instability in the laser-microresonator detuning. With approximately 10 mW power out of the DFB laser, the total microcomb power (excluding the central pump line) out of the $Si_3N_4$ chip is around 0.4 mW, corresponding to a CW-laser-to-microcomb power conversion efficiency of 4%. With the optical power spectrum in Fig. 2a inset, a laser power of nearly 1 mW out of the chip is measured when the laser is completely off-resonance and in a free-running state. Thus the waveguide transmission is calculated as 10%. This relatively low transmission is mostly attributed to the coupling losses at the two facets of the $Si_3N_4$ chip, and the mode mismatch between the DFB laser mode and the $Si_3N_4$ inverse taper mode. The transmission can be improved by further optimization and integration of the system. Here we focus on the platicon formation dynamics and deduce that the on-chip microcomb conversion efficiency reaches 40%. Such a microcomb conversion efficiency is about 100 times higher than that of DKS microcombs in anomalous-GVD microresonators with similar FSR[11], showing the advantage of platicon microcombs in terms of power efficiency[39]. Upon photodetection of the platicon repetition rate using a fast InGaAs photodiode with a bandwidth of 45 GHz, we measure its phase noise spectrum with a phase-noise analyzer (PNA) using Welch's method from a time-sampling trace of the in-phase and quadrature components. Without any external active control to stabilize the platicon, the SSB phase noise of the repetition rate around 26.2 GHz is acquired and presented in Fig. 2c. The phase noise spectrum shows an overall slope of 30 dB/dec, with a phase noise magnitude above −40 dBc/Hz at 1 kHz Fourier offset frequency. Such a high phase noise is mainly caused by the instabilities in the laser phase noise (as can be seen in Fig. 2b) via the noise conversion mechanism that converts the

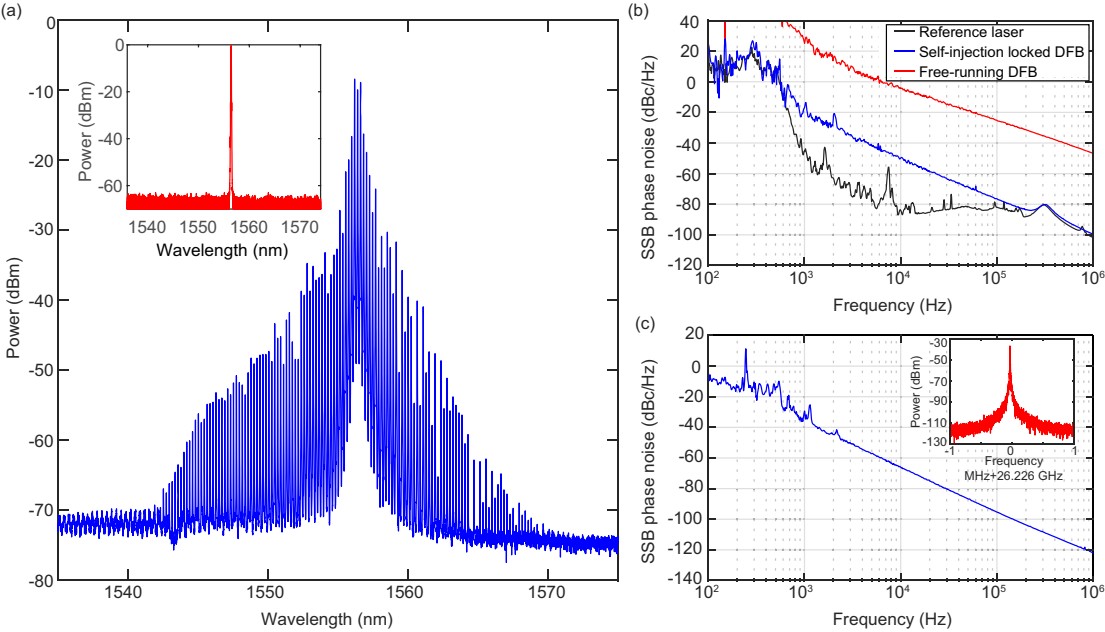

**Fig. 2 K-band microwave-repetition rate platicon microcomb. a** The optical spectrum of the coherent, single-FSR-spacing platicon microcomb with 26.2 GHz repetition rate. The inset shows the DFB laser spectrum when the laser is off-resonance and in free-running condition. **b** Comparison of SSB phase noise of free-running DFB (red) and self-injection-locked DFB (blue) beating against the reference ECDL laser (black). The DFB laser is operated at 1556 nm. The reference laser's phase noise (black) is measured against an ultrastable reference laser (MenloSystems ORNS) at 1552 nm. **c** SSB phase noise of the beat signal among the platicon comb lines at 26.226 GHz, revealing the low noise regime. Inset: repetition rate beatnote signal.

laser-microresonator detuning instability into the platicon repetition rate noise[76]. Besides using a semiconductor laser with a lower intrinsic noise, we note that reducing the higher-order dispersion of the microresonator could potentially improve the repetition rate phase noise because of the decreased noise conversion coefficient.

Next, we investigate in detail the platicon formation process. We carefully adjusted the feedback phase by varying the gap distance between the DFB laser and the $Si_3N_4$ chip (see Supplementary Information for the feedback phase study). Then we tune the laser bias current (thus the free-running laser frequency) over a microresonator resonance. We observe multiple comb states and switching with varying comb line spacings—three-, one-, or two-FSR (see Fig. 3c–e). Figure 3f shows the microresonator transmission spectrum of the pump laser (red) and the generated light (filtered out pump power) (green), upon triangular sweeping of the laser bias current (blue). The microresonator transmission spectrum shows characteristic vertical edges of the pumped resonance, marking the self-injection locking range. We observe that tuning into coherent platicon states is feasible with both forward and backward scanning of the laser current, similar to the observation in anomalous-GVD microresonators as demonstrated in ref. [59]. However, we also note that the single-FSR-spacing platicon state can only be achieved in the backward scan (i.e., decreasing laser current). To obtain more insights into the comb formation dynamics, we employ a beatnote spectroscopy with an auxiliary frequency-stabilized laser, and measure the frequency tuning curve of the self-injection-locked laser. Figure 3h shows the time-frequency spectrogram when the DFB laser's natural frequency is linearly up-swept over the resonance by decreasing the laser bias current (i.e., backward tuning). First, the laser exhibits power instability, and the spectrogram shows multiple beatnote frequencies, indicating that the laser is in a dynamically unstable multi-frequency regime. As the bias current is further decreased, the laser becomes single-frequency, and the

single beatnote frequency increases linearly until, at the time of ca. 11 s, the laser is self-injection-locked to the microresonator. In the entire locked state, the laser frequency only moves by 160 MHz, which can be seen from the zoom-in examination shown in Fig. 3g. Simultaneously, the optical spectrum shows several microcomb states, including the three-, single-, and two-FSR-spacing combs that are generated sequentially (see Fig. 3c–e). Because the frequency-stabilized laser also beats with the comb teeth, when the microcomb is single-FSR-spacing, an extra beatnote frequency appears around 13.25 GHz, which is produced by beating the auxiliary laser with the comb tooth next to the pump line. In addition, the platicon repetition rate of 26.2 GHz is also detected. At 22.5 s, the laser eventually exits the locked state, as the spectrogram shows that the laser frequency abruptly jumps by nearly 13 GHz and then resumes the linear sweeping operation. Based on the beatnote spectroscopy, as indicated by the vertical double-arrow sign from 12.0 to 25.4 GHz, the estimated full range of the laser natural frequency detuning for the locked state is ca. 13.3 GHz (also see Fig. 1g). From the estimation based on the sum of forward and backward scan locking ranges[77] the full locking range is ca. 15.6 GHz range. Such a large locking range that covers half of the microresonator FSR shows that the self-injection locking is robust, which allows for the generation of multiple different platicon states, including the perfect platicon crystal states[78] in Fig. 3c. This feature may be utilized in the on-demand switching of platicon repetition rates for variable microwave synthesis[12]. In the Supplementary Information, we present extensive simulation studies on the formation of perfect platicon crystals and discuss the underlying mechanism.

**Second-order pulse intensity autocorrelation measurement.** To further examine the characteristics of the platicon states, we perform second-order intensity autocorrelation measurement on two distinct multi-platicon states. The experimental setup is

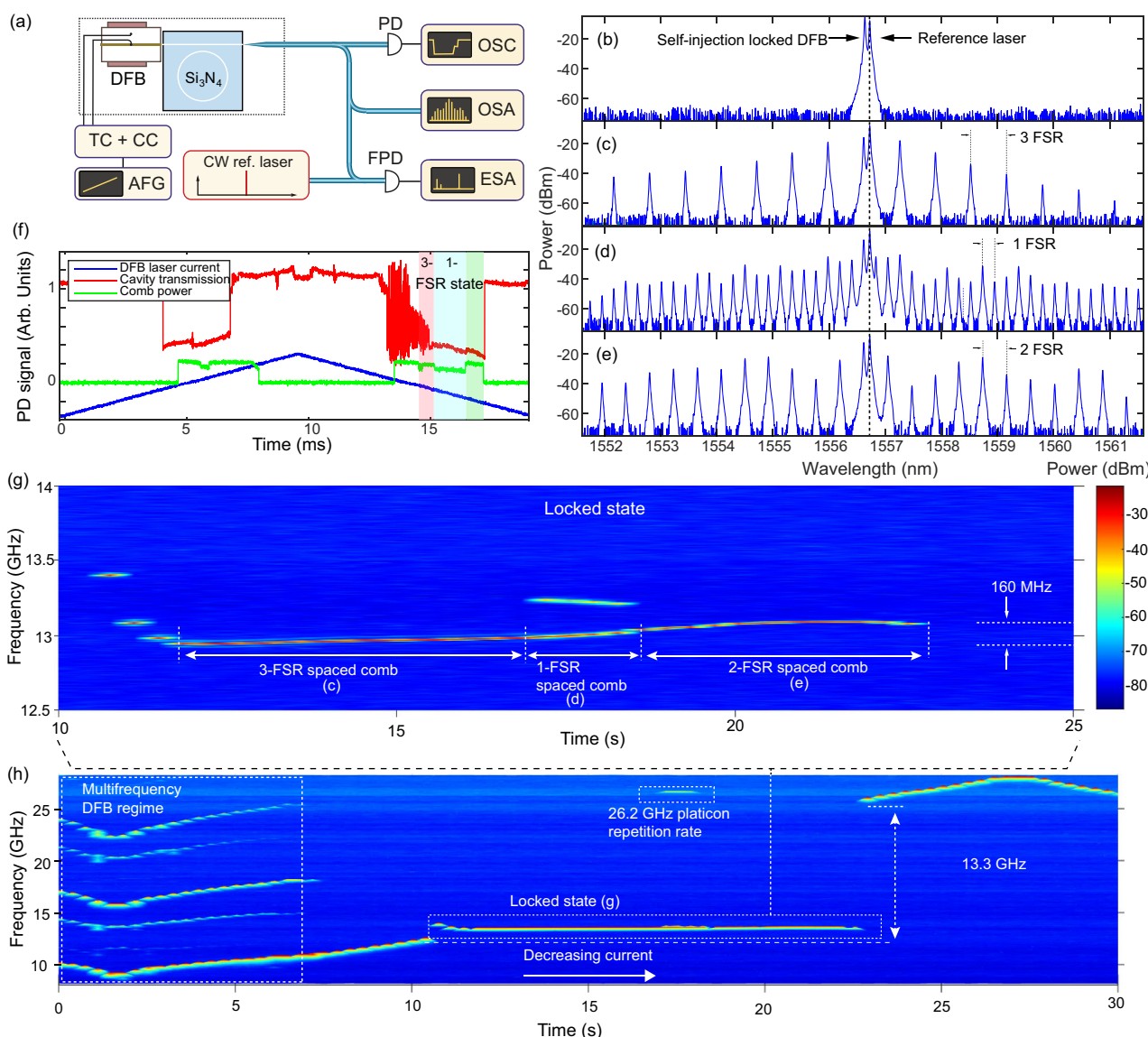

**Fig. 3 Characterization of dynamics of self-injection-locked platicon microcombs. a** Experimental setup. TC+CC: temperature and current controllers for the DFB laser. AFG: arbitrary function generator. PD, FPD: photodiode. OSC: oscilloscope. OSA: optical spectrum analyzer. ESA: electrical spectrum analyzer. **b–e** Optical spectra of different comb states upon backward tuning (decreasing laser current). The reference laser peak is at 1556.72 nm. **f** Microresonator transmission trace (red) and generated comb light (yellow) upon forward (increasing laser current) and backward scan of the DFB laser current (blue). Shaded red, blue and green areas correspond to the formation of 3-, 1-, 2- FSR spaced combs, which are delineated by the abrupt rise or fall of the comb power detected by the photodetector. **h** Beatnote spectroscopy of laser self-injection locking upon decreasing the DFB current, to reveal the laser dynamics. **g** Zoom in of (**h**) from 10 s to 25 s. The signal at 12.93 GHz corresponds to a heterodyne beat of locked laser and a reference laser. The beatnote between the first comb line and a reference laser is 13.23 GHz. The locking range is 13.3 GHz (from 12.07 to 25.37 GHz).

illustratred in Fig. 4a, and more information about the setup is presented in the Methods. The first platicon state is a dual-platicon state, whose comb spectrum displays a line spacing of one FSR (see Fig. 4b), showing that this state is not a perfect platicon crystal. The autocorrelation intensity trace presented in Fig. 4c reveals a pattern period of 38.3 ps, in agreement with the 1-FSR repetition rate of 26.2 GHz. Additionally, within each period, there are two extra intensity maxima of weaker magnitude, exhibiting the non-perfect-crystal nature of the dual-platicon state. The second platicon state measured is a perfect platicon crystal composed of four equally spaced platicons in each cavity round trip, which is shown by the comb spectrum with a 4-FSR line spacing in Fig. 4d. Figure 4e plots the measured autocorrelation trace, showing a variation period of 9.6 ps in agreement with the platicon pulse repetition rate of 104.8

GHz (4 FSRs). From this measurement, we also observe that the temporal width of the autocorrelation peak shape is around 4.3 ps. Such a temporal duration is significantly larger than the typical DKS duration of a few hundred femtoseconds. We note that due to the significant line-to-line power variations the second-order autocorrelation is dominated by very few central lines and cannot be used to faithfully determine the pulse shape. Nevertheless, together with the optical spectra, this measurement confirms the coherent structures of the platicon pulse trains and shows that the temporal separations between multiple platicons can be varied, similar to the case observed with DKS.

**Numerical simulations**. As a complementary approach to analyze the dynamics of the laser-microresonator system, a model

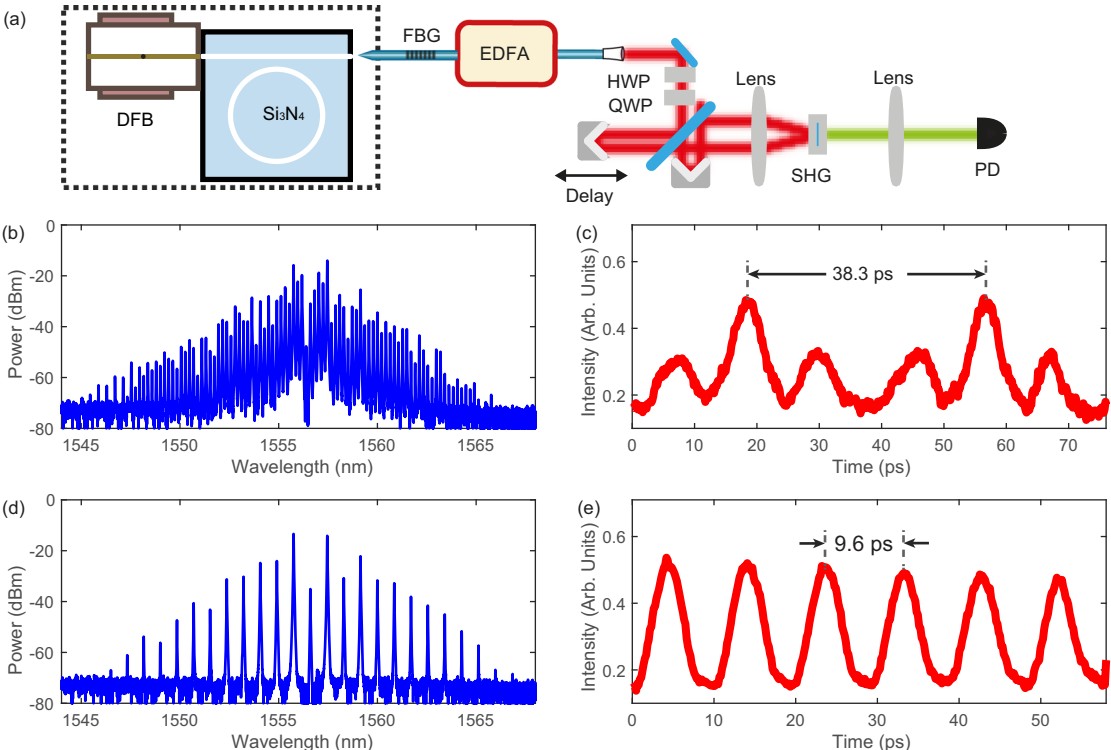

**Fig. 4 Second-order non-collinear autocorrelation measurement of platicon states. a** Experimental setup for the autocorrelation measurement. EDFA: erbium-doped fiber amplifer. HWP: half-wave-plate. QWP: quarter-wave-plate. SHG: second harmonic generation. FBG: fiber Bragg grating notch filter. **b, d** Microcomb optical spectra of a dual-platicon state and a 4-platicon perfect crystal state, respectively, with pump line suppressed by an FBG filter. Since the dual-platicon state is not a perfect crystal state, its comb line spacing is of a single FSR. For the 4-platicon perfect crystal state, the comb line spacing is 4 FSR. **c, e** The corresponding intensity autocorrelation traces of the two platicon states.

is developed to simulate the laser self-injection locking and platicon excitation in the microresonator. This model is based on coupled equations that combine the semiconductor laser rate equations[65,66] and the modified Lugiato-Lefever equation (LLE)[67,68]. We consider only backscattering in central pumped mode, our model is simplified compared to the model used in ref. [63]. The detailed description of the model and the values of parameters used in the simulations can be found in the Supplementary Information. We present the main simulation results in Fig. 5a, showing a typical microcomb generation process when the laser-cavity resonance frequency is linearly swept downwards over a microresonator resonance (see the gray dotted line in the lower panel of Fig. 5a for $(\omega_0 - \omega_{L0})/2\pi$, where $\omega_{L0}/2\pi$ is the laser-cavity resonance frequency and $\omega_0/2\pi$ is the microresonator's pumped resonance frequency). Figure 5a upper panel shows the evolution of the intracavity intensity along the azimuthal angle of the rotating frame ($\phi$), while the middle panel plots the power evolution in the pumped resonance and the generated comb power (excluding the pump power). At the beginning of the simulation, the laser detuning $(\omega_0 - \omega_L)/2\pi$ (where $\omega_L/2\pi$ is the actual laser frequency) denoted by the solid red line in the lower panel of Fig. 5a is the same as the laser cavity resonance frequency detuned from the microresonator frequency, as expected for a semiconductor laser without optical injection. As the laser cavity resonance approaches the microresonator resonance, the laser frequency abruptly shifts at 0.44 $\mu$s. Interestingly, the laser frequency starts to oscillate even before the dramatic frequency shift happens. Such an oscillation phenomenon may be caused by the so-called critical slowing down[79,80] of the system when the bifurcation point is approached. After the frequency shift, the laser detuning has a positive value of ~150 MHz, because the microresonator resonance frequency is red-shifted

due to the Kerr effect when the laser is effectively coupled into the microresonator. Shortly after, a platicon is formed by two opposite switching wavefronts. The temporal duration of the platicon gradually decreases as the intracavity power is reduced due to the increase of laser-cavity-resonance offset from the pumped resonance frequency. The platicon maintains stable existence until the laser exits from the self-injection-locked state, which shows an entire frequency range of the locked state over 2.5 GHz. We also notice the transient oscillation (with oscillation frequencies of several hundred MHz) behavior of the laser frequency when the laser state is switched, either from unlocked to self-injection-locked state, or from locked to unlocked state. Such oscillations were observed with numerical simulations in the past[54]; we attribute this to the complex coupled dynamics of two nonlinear subsystems. Oscillations in semiconductor lasers subject to optical injections have been intensively investigated in the past, and here we do not intend to study this effect in detail. Yet, one should note that oscillation at the state switching of a non-linear system is a universal phenomenon.

Figure 5b, c shows the microcomb spectrum at the time of 2.1 $\mu$s and the corresponding temporal profile of the intracavity platicon, respectively. Relatively good agreement between the simulated comb spectrum and the experimental observation is achieved. We note that the results shown here are only one representative example. The successful microcomb generation and laser self-injection locking can be obtained with varied feedback phases and laser powers. Recently the physics of stable platicon generation enabled by self-injection locking technique has been analyzed[64], showing that the self-injection locking can lock the laser at a detuning usually not stably accessible with a conventional laser scanning pumping scheme. In addition, we also simulate the formation of the perfect platicon crystals

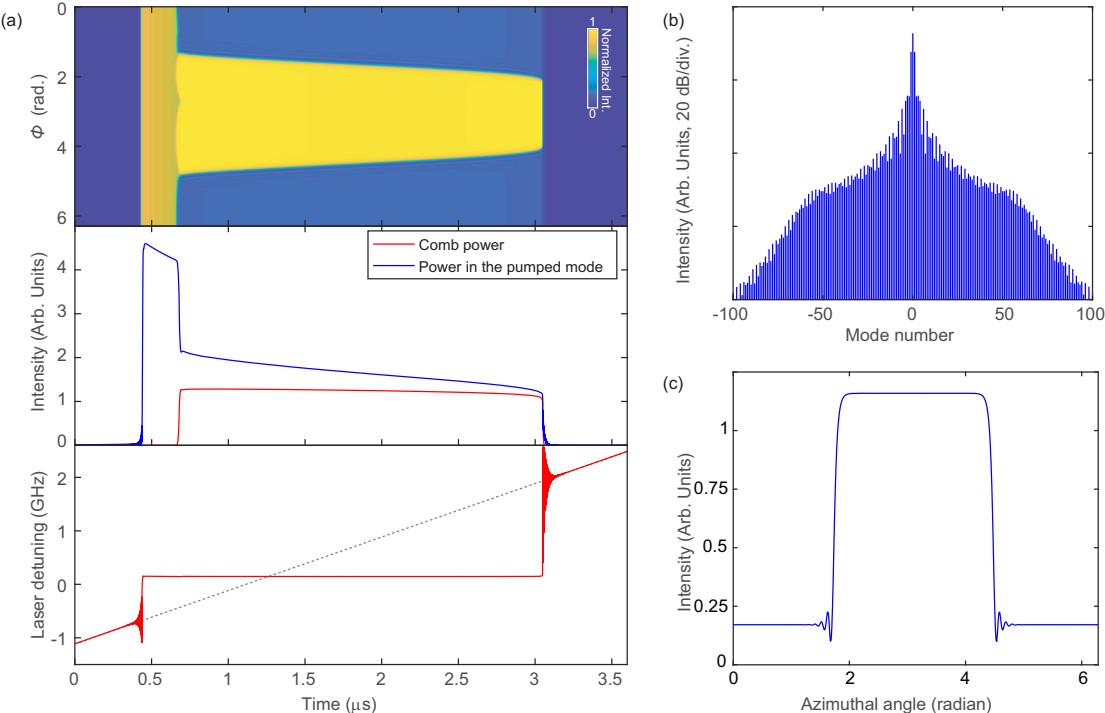

**Fig. 5 Numerical simulations of platicon microcomb generation with laser self-injection locking. a** Dynamic evolution of the microresonator intracavity intensity (upper panel), the intracavity comb power and the power in the pumped resonance (middle panel), and the laser detuning of the semiconductor laser with respect to the cold microresonator resonance (bottom panel) as the laser cavity resonance frequency is swept downwards across the microresonator resonance. The dashed line in the bottom panel indicates the lasing frequency trajectory if the optical feedback for self-injection locking is absent. **b** The platicon spectrum at the time of 2.1 $\mu$s. **c** The temporal profile of the platicon corresponding to the microcomb spectrum in (**b**).

observed experimentally. The results and the details of the simulation are included in the Supplementary Information.

## Discussion

In summary, we have demonstrated platicon microcomb generation in an integrated $Si_3N_4$ microresonator with normal GVD and strong confinement waveguides (opposite to low confinement platform[73]) using a compact semiconductor laser. Employing laser self-injection locking, variable platicon states are directly initiated via simply laser current tuning. As shown by our numerical simulations and in-depth analysis in ref. [64], the self-injection locking allows the system to enter a dynamically stable microcomb state that is otherwise unstable in the conventional pump condition, thus alleviating the limits on cost, design and fabrication.

Compared with DKS microcombs, platicon microcombs have a relatively narrow optical spectral span, which might be expanded by incorporating higher-order dispersion effect or using microresonators with strong second-order nonlinear optical susceptibility[81]. Nonetheless, platicon microcombs exhibit high energy conversion efficiency, and can be efficiently utilized at visible wavelength ranges where microresonators with anomalous GVD are scarcely achieved. As such, with the reinforcement of laser self-injection locking, we expect platicon microcombs to enable an indispensable complement to DKS microcombs for wide applications in spectroscopy and metrology.

## Methods

**Autocorrelation measurement**. To measure the intensity autocorrelation traces, the platicon comb light is split interferometrically and recombined with an adjustable delay in a barium borate crystal to generate the second harmonic signal in a non-collinear fashion. The intensity of the second harmonic signal is continuously measured as the delay is scanned. Since the fiber used to transfer the

platicons to the autocorrelation setup is non-polarization-maintaining, the traces may display intensity drifts due to the polarization change in the several-minute-long measurement. In Fig. 4c, a linear intensity drift has been subtracted from the original trace. No subtraction is performed to the trace in Fig. 4e.

## Data availability

Data used to produce the plots within this paper is available at https://doi.org/10.5281/zenodo.5809186. All other data used in this study are available from the corresponding authors upon reasonable request.

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

## Acknowledgements

This work was supported by the Air Force Office of Scientific Research (AFOSR) under Award No. FA9550- 19-1-0250, by Contract HR0011-20-2-0046 (NOVEL) from the Defense Advanced Research Projects Agency (DARPA), Microsystems Technology Office (MTO), by Swiss National Science Foundation under grant agreement No. 176563 (BRIDGE), and by the European Union H2020 research and innovation program under FET-Open grant agreement no. 863322 (TeraSlice). Y.L. acknowledges support from the EU H2020 researchand innovation program under Marie Sklodowska-Curie IF grant agreement No. 898594 (CompADC). The $Si_3N_4$ samples were fabricated in the Center of MicroNanoTechnology (CMi) at EPFL. We thank Freedom Photonics for providing the DFB laser, and Dave Kinghorn for contribution in laser packaging.

## Author contributions

G.L. and W.W. performed the experiments with assistance from M.H.A. and Y.L. J.L. and R.N.W. designed and fabricated the $Si_3N_4$ samples. J.H., J.L. and G.L. characterized the $Si_3N_4$ samples. W.W. performed the numerical simulations. L.C. and J.G. provided the DFB laser. W.W., G.L., and J.L. wrote the manuscript, with the input from others. T.J.K. and J.E.B. supervised the collaboration. Correspondence and requests for materials should be addressed to T.J.K.

## Competing interests

The authors declare no competing interests.
