## [Peer review file · Nature Communications]

REVIEWER COMMENTS

Reviewer #1 (Remarks to the Author):

In this paper, Lihachev et al. experimentally demonstrated dark soliton generation in a silicon nitride (Si₃N₄) resonator with microwave repetition rate. The generation process is initiated by self-injection locking of a DFB laser to the resonator, similar to the previous turnkey schemes. Phase noise of optical and repetition rate signals have been characterized, and the laser dynamics are monitored by beating with a reference laser. Simulation results have also been provided based on the dynamical equations. I ask the authors to analyze the data more thoroughly taking into consideration the comments below. The paper can be accepted if these points are addressed.

Major comments:

1, The authors have measured a 26.2 GHz FSR for their resonator with 900 μm radius, which is typical for a Si₃N₄ resonator. This also agrees with the comb line spacing in Fig. 2a. However, In Fig. 3e, the optical spectrum has about 12 lines between 1555 and 1560 nm, indicating the FSR marked on the plot equals about 52 GHz. As the authors did not mention using a different resonator for the two experiments, this creates a discrepancy between the different spectrums being measured.

2, Related to the previous point, in Fig. 3g, there is an extra beatnote appearing in the “1 FSR” region above the continuous beatnote in the center. The authors should explain the origin of this beatnote. If the beatnote is the result of the reference laser beating with an adjacent comb line, then this creates a discrepancy with optical spectrum as observed in Fig. 3e.

3, Numerical simulations have been presented in Fig. 4a, in which the free-running laser frequency is down-scanned towards the resonance. The appearance of a ringdown feature before the frequency jump of the laser needs to be explained. Is this a general feature of injection-locked laser being pulled into a resonance.

Minor comments:

4, How did the author delineate the different FSR states in Fig. 4f? The 5-FSR region also includes a section where the cavity transmission is oscillating. Does this mean that the 5-FSR state forms breathers at the corresponding range of free-running laser frequency?

5, There are no obvious dispersive waves in the dispersion spectrum as measured by the authors, yet multiple-FSR states can still be observed in the experiment. The authors should provide simulation results indicating that multiple-FSR states are feasible in the current system.

6, What is the statistics of back-reflection levels for different longitudinal modes?

7, It is not clear what is meant by "Lasing frequency" in Fig. 4a. Is it equivalent to the laser detuning with respect to the cold cavity?

Reviewer #2 (Remarks to the Author):

The manuscript is about experimental demonstration of solitonic pulse -- platicon -- generation at normal group velocity dispersion (GVD) by means of the self-injection locking technique (SIL). The ability to use the normal GVD microresonators significantly relaxes the material and geometry design restrictions. Furthermore, the platicons are more energy-efficient than their anomalous dispersion counterpart. However, such generation is impossible without special techniques and to date there is a limited number of them. The SIL is a novel methodic, actively studied now. Thus this topic has quite high relevance.

However, the novelty of this work should be greatly improved. The main claim of the first platicon observation in SIL regime is not entirely correct. The first demonstration was shown in the work [Wei Liang, Anatoliy A. Savchenkov, Vladimir S. Ilchenko, Danny Eliyahu, David Seidel, Andrey B. Matsko, and Lute Maleki, "Generation of a coherent near-infrared Kerr frequency comb in a monolithic microresonator with normal GVD," *Opt. Lett.* 39, 2920-2923 (2014)] in crystalline microresonators. Then the regimes and boundaries of such generation were studied numerically [Nikita M. Kondratiev, Valery E. Lobanov, Evgeny A. Lonshakov, Nikita Yu. Dmitriev, Andrey S. Voloshin, and Igor A. Bilenko, "Numerical study of solitonic pulse generation in the self-injection locking regime at normal and anomalous group velocity dispersion," *Opt. Express* 28, 38892-38906 (2020)] and on-chip demonstration was performed [Jin, W., Yang, QF., Chang, L. et al. Hertz-linewidth semiconductor lasers using CMOS-ready ultra-high-Q microresonators. *Nat. Photonics* (2021). <https://doi.org/10.1038>]. So, the presented results should be compared to the highlighted works. Authors also should provide stronger, direct evidences of platicon observation, such as autocorrelation, FROG or electro-optic sampling traces to bring out some novelty. Multiple-FSR states' origin could be probably discussed more as platicon spectra usually are single-FSR spaced. The provided dynamical spectrograms also need to be better explained.

Concerning this, unfortunately, I can not recommend this article to be published in Nature Communications in the current state.

Other major and minor issues in the manuscript arranged by line number are listed in the following:

line 23-25: The statement is incorrect. It should be noted here that self-injection locked platicon generation at normal GVD was demonstrated in crystalline microresonators [Wei Liang, Anatoliy A. Savchenkov, Vladimir S. Ilchenko, Danny Eliyahu, David Seidel, Andrey B. Matsko, and Lute Maleki, "Generation of a coherent near-infrared Kerr frequency comb in a monolithic microresonator with normal GVD," *Opt. Lett.* 39, 2920-2923 (2014)], studied theoretically [Nikita M. Kondratiev, Valery E. Lobanov, Evgeny A. Lonshakov, Nikita Yu. Dmitriev, Andrey S. Voloshin, and Igor A. Bilenko, "Numerical study of solitonic pulse generation in the self-injection locking regime at normal and anomalous group velocity dispersion," *Opt. Express* 28, 38892-38906 (2020)] and demonstrated experimentally on-chip [Jin, W., Yang, QF., Chang, L. et al. Hertz-linewidth semiconductor lasers using CMOS-ready ultra-high-Q microresonators. *Nat. Photonics* (2021). <https://doi.org/10.1038>].

line 76: The citations for the modulational excitation looks missing. This technique was proposed and studied in [Valery E. Lobanov et al 2015 EPL112 54008] and [Valery E. Lobanov, et. al. *Phys. Rev. A* 100, 013807 – Published 3 July 2019] and should be cited to be comprehensive. It should be noted that not only amplitude modulation is possible. The possibility of using the phase modulation on subharmonics was shown in the second mentioned work. For the experimental demonstration of the amplitude modulation method [H. Liu, S. - Huang, J. Yang, M. Yu, D. - Kwong, and C. W. Wong, "Bright square pulse generation by pump modulation in a normal GVD microresonator," in *Conference on Lasers and Electro-Optics, OSA Technical Digest (online) (Optical Society of America, 2017)*, paper FTu3D.3.] or [H. Liu, W. Wang, S. - Huang, M. Yu, D. - Kwong, and C. W. Wong, "Extended access to self-disciplined platicon generation in normal dispersion regime via single FSR intensity-modulated pump," in *Conference on Lasers and Electro-Optics, OSA Technical Digest (Optical Society of America, 2020)*, paper SF2B.2.] should be cited.

Figure 1a: The term "PSD" (power spectral density) is usually used for noises, while here the term "power" or "optical spectrum" looks more appropriate.

Figure 1g: It is written that the panel shows the nonlinear self injection-locking. However, the nonlinear model predicts the vertical shift and deformation of the locked part of the curve, so this picture looks like a linear case.

line 119: It would be good to provide also measured value of coupling, or at least the loaded linewidth.

line 141-143: Was the central line (pump) subtracted from the output power? Could you please provide more details on the efficiency calculation?

Figure 3g is neither addressed well in the caption nor explained in the text. What is the difference between it and the Figure 3h? Is it the locked state zoom-in? Please synchronize the time values (horizontal axis) for more clarity. The vertical scale should also be changed for the picture not to look like a plain horizontal line. The arrows with "x FSR" captions are misleading as look like the region width estimations. Is the "signal at 13 GHz" in the caption referred to the main long horizontal line or to the short horizontal line in the middle?

Figure 3h: the arrow "decreasing current", pointing to the left is misleading as the time is going to the right. So was the current just increasing in this realization? What is the vertical white line between 25s and 30s?

line 172: Please, check the estimated locking range. The fact that the unlocked-to-locked frequency jump, the heterodyne beat of SIL laser and a reference laser and the locking range all have the same value is misleading.

line 173: It is said that Figure 1(g) shows the analytical estimation of the frequency tuning curve based on the nonlinear self-injection using the parameters in the experiment. However the Figure 1(g) does not look like the tuning curve in nonlinear case.

line 178: Which frequency the laser cavity offset is calculated from? It is better to write out a full formula as the sign is also important.

Equations (1)-(4): The origin of the equations should be shown. Either their derivation or proper citation should be provided. As they differ from those in [Nikita M. Kondratiev, Valery E. Lobanov, Evgeny A. Lonshakov, Nikita Yu. Dmitriev, Andrey S. Voloshin, and Igor A. Bilenko, "Numerical study of solitonic pulse generation in the self-injection locking regime at normal and anomalous group velocity dispersion," *Opt. Express* 28, 38892-38906 (2020)] and [Jin, W., Yang, QF., Chang, L. et al. Hertz-linewidth semiconductor lasers using CMOS-ready ultra-high-Q microresonators. *Nat. Photonics* (2021). <https://doi.org/10.1038>] a comparison should probably be made.

line 181: The word "field" is probably missed after (cw).

line 182: The ϕ is not an azimuthal angle, but an angle in a frame, rotating with D1 frequency.

line 181,184,191-192: How are κ_r , κ_{sc} , κ_{ex} , κ_{Lo} connected with defined previously κ_0 and β ?

line 192-195: It is unclear, what was made to to increase the computation efficiency. How is the definition of the fields connected with the computational efficiency?

line 196 how was the parameter η estimated?

line 200 "is" should be replaced with "are"

line 205 the word "and" is probably to be removed

line 205-206 is it the laser power or the intra-cavity power reduced?

Figure 4a: In the upper panel there is an artefact white line from the upper left to the bottom right corner. That should be fixed. The way of the instantaneous frequency calculation in the lower lower panel should be described. Is it possible to compare the modelling (Fig. 4a, bottom), analytics (Fig 1g) and experiment (Fig. 3h) in the single figure for more clarity?

line 212-213: It is well known that the SIL is highly dependent on the locking phase. How was the phase variation performed and which values were tested? This should be described in more details.

line 227-228: Are these parameters the same as those measured in the main text (line 199, 122)?

line 229: the dimension of g should be rad/s.

We appreciate the careful review by the reviewers and have modified the manuscript in accordance with their suggestions. Besides the revisions we made to the main manuscript, we have also created a Supplementary Information file that includes new data and analysis. Here, we present a point-by-point reply (in **blue**) to the reviewers' comments (in **black**), as well as the action taken (in **red**).

Reviewer #1

In this paper, Lihachev et al. experimentally demonstrated dark soliton generation in a silicon nitride (Si₃N₄) resonator with microwave repetition rate. The generation process is initiated by self-injection locking of a DFB laser to the resonator, similar to the previous turnkey schemes. Phase noise of optical and repetition rate signals have been characterized, and the laser dynamics are monitored by beating with a reference laser. Simulation results have also been provided based on the dynamical equations. I ask the authors to analyze the data more thoroughly taking into consideration the comments below. The paper can be accepted if these points are addressed.

We thank Reviewer #1 for the positive evaluation of our work.

Action taken: For the revised manuscript, we not only make revisions to the original text but also include new data, both experimental and numerical, in the main text and the newly added Supplementary Information. All the issues brought up by the reviewers have been addressed in the revised manuscript.

Major comments:

Q1. The authors have measured a 26.2 GHz FSR for their resonator with 900 μm radius, which is typical for a Si₃N₄ resonator. This also agrees with the comb line spacing in Fig. 2a. However, In Fig. 3e, the optical spectrum has about 12 lines between 1555 and 1560 nm, indicating the FSR marked on the plot equals about 52 GHz. As the authors did not mention using a different resonator for the two experiments, this creates a discrepancy between the different spectrums being measured.

We thank the reviewer for carefully reading our manuscript. The reviewer is correct in pointing out the discrepancies. We apologize that in the initial submission, the comb line separations were inconsistent with the text. In fact, we noticed this mistake and have already corrected it in our arXiv version (arXiv:2103.07795, v2 updated on March 16th). We have corrected Fig. 3 in the revised manuscript.

Action taken: A new Fig. 3 containing the correct comb spectra of 3-, 1-, 2-FSR have been added to replace the spectra in the original submission. Corresponding revisions to the captions and text have been made too.

Q2. Related to the previous point, in Fig. 3g, there is an extra beatnote appearing in the "1 FSR" region above the continuous beatnote in the center. The authors should explain the origin of this beatnote. If the beatnote is the result of the reference laser beating with an adjacent comb line, then this creates a discrepancy with optical spectrum as observed in Fig. 3e.

As we stated in the last question, the correct comb spectra have replaced the original spectra. The reviewer is right that the extra beatnote results from the reference laser beating with an adjacent comb line.

Action taken:

Action taken: The comb spectra have been corrected, showing the correspondence between the comb spectra in Fig.3(b-e) and the beat note spectrogram evolution in Fig.3(g-h). The cause for the extra beatnote is explained in the corresponding text.

Q3. Numerical simulations have been presented in Fig. 4a, in which the free-running laser frequency is down-scanned towards the resonance. The appearance of a ringdown feature before the frequency jump of the laser needs to be explained. Is this a general feature of injection-locked laser being pulled into a resonance.

When the laser-microresonator state changes from unlocked to self-injection locked, transient response of the laser frequency and amplitude almost always happens due to the nature of the coupling of multiple physical states that have nonlinear responses and different response times. Since the laser frequency is suddenly changed and the feedback optical power and phase would also exhibit complex dynamical response, we believe that the transient oscillations that appear in the simulation are caused by the mutual interaction between the ringdown effect of the microresonator (with Kerr nonlinear response) and the oscillation of the semiconductor laser (in both carrier number and light field amplitude and phase). Although experimentally observing such transient behavior in the time domain would be too challenging due to the short timescale, we believe that the simulation model captures the feature qualitatively. Here an extensive study on this phenomenon is out of the scope. Yet, we note that in a theoretical publication studying self-injection locked lasers [Kondratiev, N. M. et al. Self-injection locking of a laser diode to a high-Q WGM microresonator. *Optics Express* 25, 28167–28178 (2017)], a similar feature was observed in the numerical simulation.

Action taken: In the revised manuscript, when explaining the results of Fig. 4 (now Fig.5 in the revised manuscript), the mechanisms responsible for the transient feature are presented in the text. We also added the OE reference to corroborate our results.

Minor comments:

Q4, How did the author delineate the different FSR states in Fig. 4f? The 5-FSR region also includes a section where the cavity transmission is oscillating. Does this mean that the 5-FSR state forms breathers at the corresponding range of free-running laser frequency?

We assume that the reviewer was referring to Fig.3f. We used different laser sweeping speeds, and we found that the transmission and comb power spectra were almost identical. Therefore, we believe that the comb state evolution in Fig.3(g,h) is the same as those in Fig.3f, despite that they were two separate takes (at 50 Hz and 20 mHz laser scanning rate, respectively). To delineate the different platicon states, we observed that the comb power in Fig.3f (green trace) shows an abrupt increase (step up) or decrease (step down), which are in close agreement with the different states in Fig.3 (c,d,e). Therefore, we use the abrupt change of the comb power as the separation lines for different states. As to the observed transmission oscillation, we suspect it was caused by the instability of the semiconductor laser when the laser is subject to specific optical feedback. Such instability in semiconductor lasers with feedback has been widely observed and studied by the semiconductor laser community. At the same time, the platicon state might also demonstrate unstable powers with particular pump and detuning conditions. In some experiments, we also observed sidebands on the platicon repetition rate signal, which could be attributed to the breathing frequency – a very similar observation to breathers in a bright soliton regime. However, we feel that a thorough investigation into this particular phenomenon is beyond the scope of this work, so we only provide qualitatively reasonable explanations in the manuscript.

Action taken: In the revised manuscript, in the caption of Fig.3, we explain how we delineate the different platicon states based on the abrupt changes in comb power evolution. In the text, we describe the laser instability regime before the self-injection-locked state is reached.

Q5, There are no obvious dispersive waves in the dispersion spectrum as measured by the authors, yet multiple-FSR states can still be observed in the experiment. The authors should provide simulation results indicating that multiple-FSR states are feasible in the current system.

We thank the reviewer for pointing out this issue about the absence of obvious dispersive waves. After we received this review report, we performed extensive simulations. We found that even with a small frequency deviation from the smooth D_2 -only dispersion profile, equally spaced multi-platicon state, or "platicon crystals," can be formed due to the dispersion perturbations. In a real experiment, such dispersion perturbations are inevitable because of the multi-mode nature of the microresonator.

Action taken: In the newly added Supplementary Information, we added new simulations that include small dispersion deviations and that could reproduce the multi-FSR-spaced platicon crystal states. The simulation procedure and details are elaborated in the SI "Simulation of the formation of perfect platicon crystal states"

section.

Q6, What is the statistics of back-reflection levels for different longitudinal modes?

We add characterization data for the sample performed with frequency comb calibrated diode laser spectroscopy technique (see Supplementary Figure 5 in the Supplementary Information and Figure R1 below). We provide broadband measurements of the transmission and reflection spectra and statistics of the backreflection, intrinsic cavity loss (κ_0), and bus waveguide coupling rate (κ_{ex}). Transmission trace calibration includes input and output lensed fiber losses. From the reflection trace, no clear dependence of backreflection rate is visible. The histogram shows that half of the resonances have a backreflection of 0.05 in power – the level of chip facets reflection. Still, more than 50 resonances are available with 0.1-0.15 backreflection, which allows increasing the self-injection locking range.

Figure R1. Linear spectroscopy of photonic chip. (a) Frequency-dependent microresonator intrinsic loss κ_0 and bus waveguide coupling κ_{ex} . (b) Histogram of backreflection values from the cavity resonances. (c) Transmission (blue) and (d) reflection (red) of the microresonator with an FSR of 26.2 GHz.

Action taken: We added the characterization figure in the SI to provide additional information on the microresonator samples.

Q7, It is not clear what is meant by "Lasing frequency" in Fig. 4a. Is it equivalent to the laser detuning with respect to the cold cavity?

We thank the reviewer for this comment. The "lasing frequency" is the DFB laser's output frequency, and indeed that it is equivalent to laser detuning with respect to the cold cavity resonance. In Fig. 4a (now Fig.5a) bottom, we agree that the label is better to be "Laser detuning."

Action taken: The y-label of Fig. 4a (now the Fig.5a in the revised manuscript) bottom is changed to "Laser detuning". The figure caption has also been revised.

Reviewer #2

The manuscript is about experimental demonstration of solitonic pulse -- platicon -- generation at normal group velocity dispersion (GVD) by means of the self-injection locking technique (SIL). The ability to use the normal GVD microresonators significantly relaxes the material and geometry design restrictions. Furthermore, the platicons are more energy-efficient than their anomalous dispersion counterpart. However, such generation is impossible without special techniques and to date there is a limited number of them. The SIL is a novel methodic, actively studied now. Thus this topic has quite high relevance.

Q8. However, the novelty of this work should be greatly improved. The main claim of the first platicon observation in SIL regime is not entirely correct. The first demonstration was shown in the work [Wei Liang, Anatoliy A. Savchenkov, Vladimir S. Ilchenko, Danny Eliyahu, David Seidel, Andrey B. Matsko, and Lute Maleki, "Generation of a coherent near-infrared Kerr frequency comb in a monolithic microresonator with normal GVD," *Opt. Lett.* 39, 2920-2923 (2014)] in crystalline microresonators. Then the regimes and boundaries of such generation were studied numerically [Nikita M. Kondratiev, Valery E. Lobanov, Evgeny A. Lonshakov, Nikita Yu. Dmitriev, Andrey S. Voloshin, and Igor A. Bilenko, "Numerical study of solitonic pulse generation in the self-injection locking regime at normal and anomalous group velocity dispersion," *Opt. Express* 28, 38892-38906 (2020)] and on-chip demonstration was performed [Jin, W., Yang, QF., Chang, L. et al. Hertz-linewidth semiconductor lasers using CMOS-ready ultra-high-Q microresonators. *Nat. Photonics* (2021). <https://doi.org/10.1038>]. So, the presented results should be compared to the highlighted works. Authors also should provide stronger, direct evidences of platicon observation, such as autocorrelation, FROG or electro-optic sampling traces to bring out some novelty. Multiple-FSR states' origin could be probably discussed more as platicon spectra usually are single-FSR spaced. The provided dynamical spectrograms also need to be better explained. Concerning this, unfortunately, I can not recommend this article to be published in *Nature Communications* in the current state.

We sincerely appreciate the reviewer's comments and suggestions that allow us to improve the quality of the manuscript. After receiving the review reports, we revised the work presentation and carried out further experimental and numerical investigations, generating new data sets that make the manuscript more comprehensive and contain novelties that have not been shown before. In particular, based on the reviewer's suggestions, we carried out autocorrelation measurement on two different platicon states, one of which is a perfect platicon crystal while the other is a non-crystal dual-platicon state. This measurement reveals the temporal profiles of the platicon states, especially the equally spaced perfect platicon crystal that multiply the platicon repetition rate. We also use a numerical simulation method to investigate the mechanism responsible for forming platicon crystal states with varied repetition rates. In addition, we investigate the impact of varying the feedback phase of the back-scattered laser light for self-injection locking, and we analyze the transient behavior of the laser during the state transition. We believe that this revised manuscript has been significantly improved with novel results delivered in the form of both experimental measurements and numerical data, and we hope the reviewer finds our new results interesting.

Action taken: Besides many revisions on the presentation, several aspects have been revised/extended with new data, including:

1. The autocorrelation measurement of platicon and perfect platicon crystal states (added in the main manuscript).
2. The numerical simulation of the formation of perfect platicon crystal states with varied composite platicon numbers (added in the Supplementary Information).
3. The qualitative depiction of the laser-microresonator dynamical states with varied feedback optical phase and laser-cold-cavity detuning, which is based on extensive simulation results. Complementary experimental data are also provided (added in the Supplementary Information).
4. A significant portion of the main manuscript texts has been revised to give readers a better understanding of the field, especially the prior works that are highly relevant to this work. These prior works are either on using microresonators for laser self-injection locking or on the generation of platicon microcombs.

Other major and minor issues in the manuscript arranged by line number are listed in the following:

Q9. line 23-25: The statement is incorrect. It should be noted here that self-injection locked platicon generation at normal GVD was demonstrated in crystalline microresonators [Wei Liang, Anatoliy A. Savchenkov, Vladimir S. Ilchenko, Danny Eliyahu, David Seidel, Andrey B. Matsko, and Lute Maleki, "Generation of a coherent near-infrared Kerr frequency comb in a monolithic microresonator with normal GVD," *Opt. Lett.* 39, 2920-2923 (2014)], studied theoretically [Nikita M. Kondratiev, Valery E. Lobanov, Evgeny A. Lonshakov, Nikita Yu. Dmitriev, Andrey S. Voloshin, and Igor A. Bilenko, "Numerical study of solitonic pulse generation in the self-injection locking regime at normal and anomalous group velocity dispersion," *Opt. Express* 28, 38892-38906 (2020)] and demonstrated experimentally on-chip [Jin, W., Yang, QF., Chang, L. et al. Hertz-linewidth semiconductor lasers using CMOS-ready ultra-high-Q microresonators. *Nat. Photonics* (2021). <https://doi.org/10.1038>].

We thank the reviewer for the comment and for providing us the references. Corresponding revisions have been incorporated in the updated manuscript.

Action taken: In the introduction of the revised manuscript, we present the previous efforts that include all the references mentioned by the reviewer.

Q10. line 76: The citations for the modulational excitation looks missing. This technique was proposed and studied in [Valery E. Lobanov et al 2015 EPL112 54008] and [Valery E. Lobanov, et. al. *Phys. Rev. A* 100, 013807 – Published 3 July 2019] and should be cited to be comprehensive. It should be noted that not only amplitude modulation is possible. The possibility of using the phase modulation on subharmonics was shown in the second mentioned work. For the experimental demonstration of the amplitude modulation method [H. Liu, S. -. Huang, J. Yang, M. Yu, D. -. Kwong, and C. W. Wong, "Bright square pulse generation by pump modulation in a normal GVD microresonator," in Conference on Lasers and Electro-Optics, OSA Technical Digest (online) (Optical Society of America, 2017), paper FTu3D.3.] or [H. Liu, W. Wang, S. -. Huang, M. Yu, D. -. Kwong, and C. W. Wong, "Extended access to self-disciplined platicon generation in normal dispersion regime via single FSR intensity-modulated pump," in Conference on Lasers and Electro-Optics, OSA Technical Digest (Optical Society of America, 2020), paper SF2B.2.] should be cited.

We sincerely thank the reviewer for pointing out these important references to us. Indeed these works show the earlier efforts in using modulation schemes to pump platicons.

Action taken: In the revised manuscript, when the platicon generation is introduced, the method based on the modulation (both AM and PM) excitation is introduced before the introduction of using self-injection locking. All the relevant references of the previous achievements recommended by the reviewer have been added.

Q11. Figure 1a: The term "PSD" (power spectral density) is usually used for noises, while here the term "power" or "optical spectrum" looks more appropriate.

We thank the reviewer for the suggestion. We have corrected the label in Fig. 1a.

Action taken: We have changed the label of "PSD" to "Power" in Fig. 1a.

Q12. Figure 1g: It is written that the panel shows the nonlinear self injection-locking. However, the nonlinear model predicts the vertical shift and deformation of the locked part of the curve, so this picture looks like a linear case.

We thank the reviewer for pointing out this issue. Indeed, the figure here, for qualitative illustration only, is of the linear case. Although nonlinearity is more or less always present, the figure of the linear scenario simplifies the situation, allowing the readers to understand the key principle of the self-injection locking mechanism.

Action taken: We have changed the figure caption "*Simplified analytical estimation of laser frequency tuning curve based on the model of linear self injection-locking*" in Fig. 1(g). In the text, we also cite the reference for the nonlinear case and briefly explain the difference.

Q13. line 119: It would be good to provide also measured value of coupling, or at least the loaded linewidth.

We present additional chip characterization data the newly added Supplementary Information (see “Characterization of the photonic chip” section).

Action taken: In the newly added figure in the last section of the Supplementary Information, we show frequency-dependent microresonator intrinsic loss κ_0 (corresponding to the histogram in Fig. 1 of the main manuscript) and bus waveguide coupling κ_{ex} . We also add a histogram of backreflection from the resonances.

Q14. line 141-143: Was the central line (pump) subtracted from the output power? Could you please provide more details on the efficiency calculation?

In the initial manuscript, we used the ratio of the full microcomb power and the free-running laser power measured in the output lensed fiber as conversion efficiency metrics.

In the revised manuscript, we rewrite the microcomb conversion efficiency part to follow the microcomb community convention. We use the comb spectrum shown in Fig.2(a). To calculate pump-to-microcomb conversion efficiency, we measured optical power out of the photonic chip (without FBG filter, using OSA) when the laser is either in free-running (see the inset of Fig.2 (a)) or the self-injection locked state with microcomb generated (see the main data in Fig.2 (a)). We compare the microcomb power (excluding the pump line) and the laser power and calculate an on-chip conversion efficiency of 40%. We also measure the power directly out of the laser chip (in free space) and estimate a transmission loss of 10 dB that is caused by the loss at the chip facets and the imperfect mode matching between the laser output beam and the mode of the Si₃N₄ tapered waveguide. Taking this loss into account, the efficiency is estimated to be ~4%.

We also note that most of the power is in neighboring comb lines, as clearly visible from Fig.2(a). For other comb states, we calculated the on-chip conversion efficiency of 24% and 19%.

Action taken: In the revised manuscript, the microcomb conversion efficiency, either on-chip or the actual value, is presented in the text. The detailed calculation is clearly explained. The potential solution to increase the actual conversion efficiency is discussed.

Q15. Figure 3g is neither addressed well in the caption nor explained in the text. What is the difference between it and the Figure 3h? Is it the locked state zoom-in? Please synchronize the time values (horizontal axis) for more clarity. The vertical scale should also be changed for the picture not to look like a plain horizontal line. The arrows with "x FSR" captions are misleading as look like the region width estimations. Is the "signal at 13 GHz" in the caption referred to the main long horizontal line or to the short horizontal line in the middle?

We've reworked Figure 3 according to the reviewer's suggestions. Figure 3g is indeed a zoom-in of the locked state from Figure 3h. It shows the beatnote spectrogram between the comb (mainly the pumped mode) and the reference laser. We re-labeled the horizontal arrows with "3-FSR spaced comb (c)", "1-FSR spaced comb (c)", "2-FSR spaced comb (c)" to separate detuning regions for different comb states. "Signal at 13 GHz" in the caption, indeed, refers to the main long horizontal line. We rephrase the caption stating more precise values of the beat frequencies. We would like to keep the vertical axis in Figure 3g at a current scale to show that the detuning change in the SIL regime is small, of an order of few κ_0 . We align the time axis in Figure 3h to start from 0s and re-align the zoom-in Figure3 (g) time axis for clarity.

Action taken: We updated Figure 3 (g, h), and the captions based on the reviewer's comments. The evolution of the comb states is explained more clearly. Several signals, not only the beatnote between the reference laser and the pumped line but also the beatnote caused by the adjacent comb tooth and the platicon repetition rate of the single-FSR-spaced comb, are presented and explained in the main text.

Q16. Figure 3h: the arrow "decreasing current", pointing to the left is misleading as the time is going to the right. So was the current just increasing in this realization? What is the vertical white line between 25s and 30s?

We thank the reviewer for pointing out this discrepancy. We redraw the figure, and now the time axis and arrow are pointing to the right. The DFB current was decreasing over time, as seen from Figure 3f for the marked regions of 3-, 1-, 2- FSR spaced comb.

The vertical line between ~25s and ~30s (of the initial Figure 3h) is the region where our DFB emits in a multifrequency regime with optical lines separated 2.5 GHz – 4 GHz depending on the diode current. We observe such a regime in most of our measurements at high DFB currents (370-380 mA) and a small distance between the laser chip and Si₃N₄ chip (< 4 μm). However, in all presented optical spectra of platicons, the DFB was operating in a single frequency regime. Similar laser behavior was observed previously in the same butt-coupling configuration to Si₃N₄ photonic chip but with a different type of DFB [Voloshin, A.S. et al. Dynamics of soliton self-injection locking in optical microresonators. Nat Commun 12, 235 (2021)].

Action taken: We updated Figure 3(h). The main text that describes this figure is also revised to provide clear information about the figure.

Q17. line 172: Please, check the estimated locking range. The fact that the unlocked-to-locked frequency jump, the heterodyne beat of SIL laser and a reference laser and the locking range all have the same value is misleading.

We rephrase the text to add precise values. The heterodyne beat of SIL laser and a reference laser is from 12.93 GHz to 13.09 GHz (depending on the diode current). The unlocked-to-locked-to-unlocked frequency jump is 13.3 GHz (from 25.37 GHz to 12.07 GHz) which we call the locking range (in the linear SIL model). The beat between comb line + 1 and a reference laser is at 13.23 GHz.

Action taken: We added more precise values for the self-injection locking range and the beatnote frequencies in the main text.

Q18. line 173: It is said that Figure 1(g) shows the analytical estimation of the frequency tuning curve based on the nonlinear self-injection using the parameters in the experiment. However the Figure 1(g) does not look like the tuning curve in nonlinear case.

We note that this comment is related to Q12. Indeed the Fig. 1g is "**linear** self injection-locking," not "nonlinear." We have corrected this.

Action taken: We have changed the figure caption "*Simplified analytical estimation of laser frequency tuning curve based on the model of **linear** self injection-locking*" in Fig. 1g. In the main text, we also explain that for simplicity, the Kerr nonlinear frequency shift of the microresonator is not considered in this qualitative depiction. In the revised text, we mention the more accurate nonlinear tuning curve that is studied in ref.22.

Q19. line 178: Which frequency the laser cavity offset is calculated from? It is better to write out a full formula as the sign is also important.

The laser cavity offset is $f_0 - f_{L0}$, where f_0 is the microresonator resonance frequency, and f_{L0} is the laser cavity resonance frequency. In the simulation, the laser cavity resonance frequency is swept over a microresonator resonance. Therefore this offset frequency is linearly tuned, which essentially corresponds to the grey dashed line in the lower panel of Fig.5(a).

Action taken: In the main text, the expressions of the laser cavity detuning and the laser detuning are presented. The full description of the laser-microresonator system model is now presented in the newly added Supplementary Information.

Q20. Equations (1)-(4): The origin of the equations should be shown. Either their derivation or proper citation should be provided. As they differ from those in [Nikita M. Kondratiev, Valery E. Lobanov, Evgeny A. Lonshakov, Nikita Yu. Dmitriev, Andrey S. Voloshin, and Igor A. Bilenko, "Numerical study of solitonic pulse generation in the self-injection locking regime at normal and anomalous group velocity dispersion," Opt. Express 28, 38892-38906 (2020)] and [Jin, W., Yang, QF., Chang, L. et al. Hertz-linewidth semiconductor lasers using CMOS-ready ultra-high-Q microresonators. Nat. Photonics (2021). <https://doi.org/10.1038>] a comparison should probably be made.

In the revised manuscript, the coupled equations for the simulation are now presented in the Supplementary Information. We note that the laser rate equations are somewhat different from those in recent publications,

although this minor difference does not cause any conflict in explaining the dominant physical mechanism. The laser rate equations we used have been extensively used in theory and simulations in earlier papers.

Action taken: In the revised main manuscript and the Supplementary Information, several references have been added for the model based on laser rate equations. We also briefly discuss the difference between our model and those in earlier works in the Supplementary Information.

Q21. line 181: The word "field" is probably missed after (cw).

We agree with the reviewer and have revised the sentence accordingly.

Action taken: We have changed the words "*the clockwise (CW) field.*"

Q22. line 182: The φ is not an azimuthal angle, but an angle in a frame, rotating with D1 frequency.

We sincerely thank the reviewer for pointing out this issue. The reviewer is correct that the angle is not the azimuthal angle of the resonator but an angle of the rotating frame.

Action taken: In the revised manuscript, we changed the description to "the azimuthal angle along the circumference of the rotating frame."

Q23. line 181,184,191-192: How are κ_r , κ_{sc} , κ_{ex} , κ_{Lo} connected with defined previously κ_0 and β ?

The simulation model is for qualitatively reproducing the experimental results because many parameters such as the laser gain and the volume are not accurately known. As such, we do not intend to use the simulation to achieve highly quantitative agreement with the experimental observations. Having said that, we note that we choose values of the parameters reasonably, either close to the experimentally measured values (such as the coupling rates and the decay rate of the microresonator), or using values that are within a reasonable range that are accepted and used in literature (such as the laser cavity decay rate and the alpha factor).

Action taken: The definition of κ_r , κ_{sc} , κ_{ex} , κ_{Lo} are presented, and their values are stated in the Supplementary Information. Those microresonator coupling/decay rates are close to the experimentally measured values.

Q24. line 192-195: It is unclear, what was made to increase the computation efficiency. How is the definition of the fields connected with the computational efficiency?

In the original submission, the laser field in the laser rate equations is related to the photon density, and the counterpropagating fields in the microresonator are related to the intracavity photon number. We were trying to use photon densities for the microresonator fields too. However, the numerical rescaling resulted in slow computation speed, mostly due to the small value of the Kerr nonlinearity. After we received the review reports, we modified the form of the laser rate equations and used a laser field that is related to the total photon number. With such modification, the computation efficiency is not decreased, and the model becomes physically better described without the redundant unit conversion between different intracavity fields.

Action taken: In the revised manuscript, the numerical model is elaborated in the newly added Supplementary Information. All intracavity fields are directly related to the intracavity photon numbers, showing better clarity and coherence. The inaccurate statement on the computation efficiency is removed.

Q25. line 196 how was the parameter η estimated?

For the revised manuscript, the detailed description of the numerical model is presented in the Supplementary Information. We revised the form of the coupled equations, mostly by changing the carrier density and the laser field related to the photon density in the semiconductor laser rate equations to the total carrier number and the field related to the total photon number in the laser cavity volume. In this way, the four coupled equations are unified in terms of physical units, all dealing with intracavity photon numbers. As such, now the parameter η is eliminated, and the coupling between the laser and the microresonator is simply related to the out-couple rate of the laser cavity and the external coupling rate of the microresonator. Essentially, the inter-coupling loss,

such as the photon loss due to imperfect mode matching, should be included. However, since the semiconductor laser is a commercial one, the laser out-couple rate is not experimentally measured, and we do not know the accurate value. Yet, in the simulation, we use 4.6 GHz as the laser out-couple rate. Empirically this value is reasonable for semiconductor lasers, and it yields qualitatively good simulation results.

Several simulation parameters in the laser rate equations, including the Henry factor, the differential gain, the recombination rate, and the coupling strength (considering all the losses due to mode mismatch) between the laser and the microresonator, are not known. To accurately measuring these parameters are not trivial and thus beyond the scope of this work. The simulation is for qualitatively demonstrating the mechanism of platicon generation with self-injection-locked lasers. The values of these parameters in the simulation are either reasonable values we estimate based on literature or chosen to yield good simulation results. Since the η here essentially determines how much power is coupled into the resonator and how much power is scattered back to the laser, we found that too much power would result in instabilities and oscillations of the laser and the microcomb system. At the same time, a power level that is too weak would lead to either the absence of the platicon generation or a very small injection locking range.

Action taken: In the Supplementary Information, the coupled equations for the numerical modelling are revised, with unified physical units for laser fields in the laser cavity and the microresonator. With this revision, the parameter η is eliminated, and the value of the inter-coupling rate between the laser cavity and the microresonator is estimated empirically and chosen based on the qualitatively good simulation results.

Q26. line 200 "is" should be replaced with "are"

We thank the reviewer for pointing out this mistake and have corrected it.

Action taken: We have corrected the sentence "*The values of the parameters used for this simulation are included in the Method section.*"

Q27. line 205 the word "and" is probably to be removed

We agree with the reviewer and have revised the sentence accordingly.

Action taken: We have removed the word and broke the sentence into two.

Q28. line 205-206 is it the laser power or the intra-cavity power reduced?

Here we meant the **reduced intra-cavity power**. Although varying the laser frequency will change the laser output power, this effect should be much weaker than the intra-cavity power reduction due to the increased laser-microresonator detuning.

Action taken: We rephrased the sentence in the revised manuscript.

Q29. Figure 4a: In the upper panel there is an artefact white line from the upper left to the bottom right corner. That should be fixed. The way of the instantaneous frequency calculation in the lower lower panel should be described. Is it possible to compare the modelling (Fig. 4a, bottom), analytics (Fig 1g) and experiment (Fig. 3h) in the single figure for more clarity?

Our reply:

We sincerely thank the reviewer for the comments that help us improve the quality of the presentation. This issue was caused by the software we used to export the figure in eps format. The instantaneous laser frequency was calculated by taking the derivative of the instantaneous phase of the complex laser field derived from the numerical integration of the coupled equations. As to the separate figures for modelling, analytics, and experiment, we would like to note that the analytics based on the linear tuning curve is only for the qualitative illustration of the self-injection locking mechanism. And like we stated before, due to the complex nonlinear system, the numerical simulation is mainly qualitative, not being able to yield highly accurate values of the experimental observations. As such, we would like to keep them in different figures for different subsections, which agrees with the structure of the main text.

Action taken: The issue with the artificial white line has been removed from the updated figure. The instantaneous frequency is calculated by taking the derivative of the instantaneous phase of the complex field. This information has been added to the newly added Supplementary Information.

Q30. line 212-213: It is well known that the SIL is highly dependent on the locking phase. How was the phase variation performed and which values were tested? This should be described in more details.

We varied the phase parameter in the model from 0 to $2*\pi$ to repeat the simulation. The results are presented in Supplementary Figure 1.

Experimentally we measured cavity transmission and generated light at different locking phases, controlled by the distance between the laser and the Si_3N_4 chip. In Figure R2 of this reply, we present the cavity transmission (red), generated light (yellow) upon the linear tuning of the DFB current (blue) using different locking phases (a-h), covering 2π range. Comb formation with non-zero generated light (yellow trace) is possible only in (b-g) in a small range of locking phases. We found experimentally that platicon formation is very sensitive to the locking phase. It is visible from comparing transmission and generated light in Figure R2 (e,f,g). We changed the DFB – Si_3N_4 chip distance by ~ 150 nm from (e) to (g). Upon decreasing the DFB current, three different comb states were observed in (e), four different comb states in (f), and only two in (g). We used the locking phase as in (c) and (h) for the autocorrelation measurements.

Figure R2. Cavity transmission (red) and generated light (yellow) upon the linear tuning of the DFB current (blue) at different locking phases (a-h). Comb formation is present only in (b-g).

Action taken: In the newly added SI, we added the results of a phase sweeping simulation in Supplementary Figure 1. In Supplementary Note 3, the experimentally observed effect of different phases is presented to

corroborate the simulation results.

Q31. line 227-228: Are these parameters the same as those measured in the main text (line 199, 122)?

As we stated in the answers to earlier comments, the simulation is mainly for qualitatively reproducing the experimental results because many parameter values are not known accurately. Having said this, we carefully choose the parameter values for the simulation within reasonable ranges that are in accordance with the experimentally measured data or prior literature.

Action taken: These parameters are now defined, and their values are presented in the Supplementary Information.

Q32. line 229: the dimension of g should be rad/s.

We thank the reviewer for pointing out this typo.

Action taken: We have revised " $g=0.56 \text{ rad/s}$ "

REVIEWER COMMENTS

Reviewer #1 (Remarks to the Author):

In the revised version of the manuscript, the authors have made considerable efforts improving the paper and adding materials. While most comments have been fully addressed, I am not entirely convinced by the response to comment #3 (now marked Q3 in the response letter). As a final minor addition, I request the reviewers add text where appropriate to address the comment below. After this addition, the paper is acceptable for publication.

Comment: As pointed out by the authors (and I agree with them), laser oscillation at state switching have been “intensively investigated” and is a “universal phenomenon”. Such oscillations usually happen after the laser switches its state, i.e. when the laser tries to approach a new steady state. The reference [59] provided by the authors [Opt. Express 25, 28167 (2017)] shows a similar behavior, where ringdown features are present in all traces of Fig. 3 of [59]. However, oscillation before the laser switches its state is another entirely different problem. This would indicate that the laser is somehow able to foresee an incoming state change and steers away from the existing state in advance, which seems unconventional. No such oscillation before state switching can be observed in Fig. 3 of [59] (especially trace I, where two detuning jumps have been induced, but oscillations are present only after the second jump). For comparison, the new Fig. 5a in the manuscript features two oscillations located at approximately 0.4 us and 3.1 us, where the 0.4 us oscillation happens before state switching and no corresponding phenomena has been described in [59]. It is this appearance of the oscillations before the frequency jump that needs to be explained or commented upon.

Reviewer #2 (Remarks to the Author):

The article is greatly improved since the first review and most of the issues have been eliminated. However there is a feeling that the article lacks novelty. Probably the abstract should provide more information about new revelations made in the work. For example the feedback phase study, performed in the supplementary is not even mentioned in the main text. It would be good to clearly point out the differences and benefits of used approaches over the previous works (photonic chip design [74], numerical models [64,74] ect.) in the main text.

Several more issues should be noted

- 1) Injection locking and self-injection locking should not be confused in fig2b legend
- 2) It looks like there are wrong figure references in lines 161 and 164
- 3) From the second sentence in line 169 it seems like the laser current is tuned by changing the feedback phase.
- 4) Line 199. Such estimation of the locking range can give over- or underestimation depending on the locking phase. More correct estimation can be made by summing the results for the forward and backward scan, see "Artem E. Shitikov et.al., Microresonator and Laser Parameter Definition via Self-Injection Locking, Phys. Rev. Applied 14, 064047 DOI:10.1103/PhysRevApplied.14.064047"
- 5) line 241 the word "is" is duplicated
- 6) It would be good to add the backward wave coupling rate to the supplementary figure 5a to have the complete picture.

Finally I assume that the paper can be published after the revision is made.

We appreciate the careful review by the reviewers and have modified the manuscript in accordance with their suggestions. Here, we present a point-by-point reply (in blue) to the reviewers' comments (in black), as well as the action taken (in red).

Reviewer #1

In the revised version of the manuscript, the authors have made considerable efforts improving the paper and adding materials. While most comments have been fully addressed, I am not entirely convinced by the response to comment #3 (now marked Q3 in the response letter). As a final minor addition, I request the reviewers add text where appropriate to address the comment below. After this addition, the paper is acceptable for publication.

Q1. Comment: As pointed out by the authors (and I agree with them), laser oscillation at state switching have been “intensively investigated” and is a “universal phenomenon”. Such oscillations usually happen after the laser switches its state, i.e. when the laser tries to approach a new steady state. The reference [59] provided by the authors [Opt. Express 25, 28167 (2017)] shows a similar behavior, where ringdown features are present in all traces of Fig. 3 of [59]. However, oscillation before the laser switches its state is another entirely different problem. This would indicate that the laser is somehow able to foresee an incoming state change and steers away from the existing state in advance, which seems unconventional. No such oscillation before state switching can be observed in Fig. 3 of [59] (especially trace I, where two detuning jumps have been induced, but oscillations are present only after the second jump). For comparison, the new Fig. 5a in the manuscript features two oscillations located at approximately 0.4 us and 3.1 us, where the 0.4 us oscillation happens before state switching and no corresponding phenomena has been described in [59]. It is this appearance of the oscillations before the frequency jump that needs to be explained or commented upon.

We sincerely thank Reviewer #1 for the further comment on the oscillation phenomenon. While this phenomenon is not the focus of this work here, based on our investigation, we will try our best to provide qualitative explanations for this observation.

First, we repeat the numerical simulations presented in [59]. Based on the Eqs 20-23 in [59] and the provided parameter values, we solve the coupled equations. We use the ODE45 function in Matlab to solve the coupled derivative equations. We also repeat the same numerical integration by using different computational precision and with the Mathematica software. We obtain the same results, which allows us to conclude that computational artifacts do not cause such oscillations. In the two figures below are the simulation results. The only difference in the simulation setting is the feedback phase. We can see that these two simulation results are just like the results in Fig.3 in [59]. The oscillations only happen after the laser frequency switches to a new frequency state (here, it is the unlocked state).

Yet, when we use another arbitrarily chosen feedback phase, we obtain the simulation result plotted in the figure below. In this figure, the oscillation happens when the laser frequency just starts to switch state. Based on this part, although we cannot be entirely sure of the exact simulation method in [59], we speculate that it is

just a coincidence that the four curves in Fig.3 in [59] are with feedback phases that can only result in late oscillations.

Second, we would like to point out that the same subgroup of the authors of [59] has published another work on self-injection locking in 2020 (see Opt. Express 28, 38892 (2020)). In Fig.6 in that paper (see the copied figure below), the numerically simulated laser frequency curve (in light blue) shows oscillations (pointed by the black arrow #1) before the laser frequency switches to a new state (One should note that in that figure the laser frequency scans from right to left, as denoted by the brown arrow at the bottom). This corroborates our speculation that the results in Fig.3 in [59] do not exclude the possibility of early oscillations.

Third, to provide some insights into this phenomenon, we refer to two review papers on critical transitions: 1) Scheffer, M., Bascompte, J., Brock, W. et al. Early-warning signals for critical transitions. Nature 461, 53–59 (2009). <https://doi.org/10.1038/nature08227>; 2) Marten Scheffer et al. Anticipating Critical Transitions. Science 338, 344 (2012); DOI: 10.1126/science.1225244. According to these works, critical transitions are common in nonlinear systems. The self-injection locking system in our work is very much like a bistable system illustrated in the figure below (clipped from the first review paper).

As the laser frequency approaches the bifurcation point F_2 (which is essentially the point at which the laser frequency switches to a locked/unlocked state), a ubiquitous feature called “critical slowing down” could happen. The figure below (clipped from the second review paper) illustrates the mechanism of critical slowing down. Basically, as the switching point is approached, the system gradually loses its capability of restoring the system to stable equilibria from subtle perturbations. In our self-injection locking system, the laser cavity resonance is continuously swept. We believe that this active laser cavity resonance sweeping can be considered as a series of consecutive perturbations as the system gets closer to the bifurcation point. As such, before the laser frequency is actually switched to a new stable state, oscillations will start to grow in advance. In other words, the laser cannot be tightly locked to its original state, despite that it has not switched to a new state yet. This type of phenomenon has been used in many nonlinear systems as an early warning signal for critical state transitions.

Fig. 2. Critical slowing down as an indicator that the system has lost resilience and may therefore be tipped more easily into an alternative state. Recovery rates upon small perturbations (**C** and **E**) are slower if the basin of attraction is small (**B**) than when the attraction basin is larger (**A**). The effect of this slowing down may be measured in stochastically induced fluctuations in the state of the system (**D** and **F**) as increased variance and “memory” as reflected by lag-1 autocorrelation (**G** and **H**).

Action taken: In the section of Numerical Simulations in the newly updated manuscript, we revise the text by briefly describing the oscillations and the plausible cause of critical slowing down. References on early warnings for critical transitions are cited to support our explanations.

Reviewer #2

Q2. The article is greatly improved since the first review and most of the issues have been eliminated. However there is a feeling that the article lacks novelty. Probably the abstract should provide more information about new revelations made in the work. For example the feedback phase study, performed in the supplementary is not even mentioned in the main text. It would be good to clearly point out the differences and benefits of used approaches over the previous works (photonic chip design [74], numerical models [64,74] ect.) in the main text.

We thank the reviewer for this thoughtful comment. We mentioned the feedback phase study in the main text. The novelties have been more clearly stated in the discussion section in the updated manuscript.

Action taken: We have revised the abstract and the discussion section. Several aspects of the main text, including the platicon profiling, the numerical simulation, and the integrated chip laser excitation, have been enhanced.

Q3. Several more issues should be noted ... Finally I assume that the paper can be published after the revision is made.

We thank the reviewer for these comments. We have corrected these typos or mistakes.

1) Injection locking and self-injection locking should not be confused in fig2b legend.

Action taken: This figure legend has been corrected according to the reviewer's request.

2) It looks like there are wrong figure references in lines 161 and 164.

Action taken: This mistake has been corrected.

3) From the second sentence in line 169 it seems like the laser current is tuned by changing the feedback phase.

Action taken: This sentence has been revised to avoid misunderstanding.

“Next, we investigate in detail the platicon formation process. We carefully adjusted the feedback phase by varying the gap distance between the DFB laser and the Si₃N₄ chip (see Supplementary Information for the feedback study). Then we tune the laser bias current (thus the free-running laser frequency) over a microresonator resonance...”

4) Line 199. Such estimation of the locking range can give over- or underestimation depending on the locking phase. More correct estimation can be made by summing the results for the forward and backward scan, see "Artem E. Shitikov et.al., Microresonator and Laser Parameter Definition via Self-Injection Locking, Phys. Rev. Applied 14, 064047 DOI:10.1103/PhysRevApplied.14.064047"

Action taken: We agree with the reviewer. We revised the sentence and added a new reference.

“Based on the beatnote spectroscopy, as indicated by the vertical double-arrow sign from 12.0 to 25.4 GHz, the roughly estimated full range of the laser natural frequency detuning for the locked state is 13.3 GHz (also see Fig.1(g)). Estimation based on the sum of forward and backward scan locking ranges [Shitikov:2020] results in 15.6 GHz range.”

5) line 241 the word "is" is duplicated.

Action taken: This typo has been corrected.

6) It would be good to add the backward wave coupling rate to the supplementary figure 5a to have the complete picture.

Action taken: In the revised SI, the value of the coupling rate is explicitly stated.

REVIEWERS' COMMENTS

Reviewer #1 (Remarks to the Author):

The authors have addressed all of the points in my review and the paper is acceptable for publication.

Reviewer #2 (Remarks to the Author):

The authors addressed the technical comments (Q3 in the answers), but the actions reported in the reply Q2 of the answers are not seen in the text. The abstract and the discussion section are identical to the previous version and no highlighted changes to the text were made. Probably some intermediate version was sent to the re-review by mistake. Otherwise, the article seems lacking novelty and impact needed for the publication in Nature Group journals.

We appreciate positive reviews by the reviewers and have modified the manuscript to highlight changes we made to address concerns of Reviewer #2. Here, we present a point-by-point reply (in **blue**) to the reviewers' comments (in **black**), as well as the action taken (in **red**).

Reviewer #2

The authors addressed the technical comments (Q3 in the answers), but the actions reported in the reply Q2 of the answers are not seen in the text. The abstract and the discussion section are identical to the previous version and no highlighted changes to the text were made. Probably some intermediate version was sent to the re-review by mistake

In the revised abstract and the main text, we highlight text in blue to address Reviewer 2 question regarding new findings and studies performed in our work. We use high confinement waveguides, we study both experimentally and numerically multi-platicon regimes and state switching. We perform experimentally and numerically optical feedback study. As requested by this reviewer, we also mention the differences to [Jin et.al 2021] and [Kondratiev et al. 2020] in the main text.

Action taken: We have highlighted changes in the abstract, main text and the discussion section.